# UNLEASHING THE POTENTIAL OF CNNS FOR INTERPRETABLE FEW-SHOT LEARNING

## ABSTRACT

Convolutional neural networks (CNNs) have been generally acknowledged as one of the driving forces for the advancement of computer vision. Despite their promising performances on many tasks, CNNs still face major obstacles on the road to achieving ideal machine intelligence. One is that CNNs are complex and hard to interpret. Another is that standard CNNs require large amounts of annotated data, which is sometimes very hard to obtain, and it is desirable to be able to learn them from few examples. In this work, we address these limitations of CNNs by developing novel, simple, and interpretable models for few-shot learning. Our models are based on the idea of encoding objects in terms of visual concepts, which are interpretable visual cues represented by the feature vectors within CNNs. We first adapt the learning of visual concepts to the few-shot setting, and then uncover two key properties of feature encoding using visual concepts, which we call *category sensitivity* and *spatial pattern*. Motivated by these properties, we present two intuitive models for the problem of few-shot learning. Experiments show that our models achieve competitive performances, while being much more flexible and interpretable than alternative state-of-the-art few-shot learning methods. We conclude that using visual concepts helps expose the natural capability of CNNs for few-shot learning.

## 1 INTRODUCTION

After their debut (LeCun et al., 1998) in 1998, Convolutional Neural Networks (CNNs) have played an ever increasing role in computer vision, particularly after their triumph (Krizhevsky et al., 2012) on the ImageNet challenge (Deng et al., 2009). Some researchers have even claimed that CNNs have surpassed human-level performance (He et al., 2015), although other work suggests otherwise (Zhu et al., 2017). Recent studies also show that CNNs are vulnerable to adversarial attacks (Goodfellow et al., 2015). Nevertheless, the successes of CNNs have inspired the computer vision community to develop more sophisticated models (He et al., 2016; Szegedy et al., 2017).

But despite the impressive achievements of CNNs we only have limited insights into why CNNs are effective. The ever-increasing depth and complicated structures of CNNs makes them very difficult to interpret while the non-linear nature of CNNs makes it very hard to perform theoretical analysis. In addition, CNNs traditionally require large annotated datasets which is problematic for many real world applications. We argue that the ability to learn from a few examples, or few-shot learning, is a characteristic of human intelligence and is strongly desirable for an ideal machine learning system.

The goal of this paper is to develop an approach to few-shot learning which builds on the successes of CNNs but which is simple and easy to interpret. We start from the intuition that objects can be represented in terms of spatial patterns of parts which implies that new objects can be learned from a few examples if they are built from parts that are already known, or which can be learned from a few examples. We recall that previous researchers have argued that object parts are represented by the convolutional layers of CNNs (Zhou et al., 2015; Mahendran & Vedaldi, 2015) provided the CNNs are trained for object detection. More specifically, we will build on recent work (Wang et al., 2015) which learns a dictionary of **Visual Concepts** (VCs) from CNNs representing object parts, see Figure 1. It has been shown that these VCs can be combined to detect semantic parts (Wang et al., 2017) and, in work in preparation, can be used to represent objects using **VC-Encoding** (where

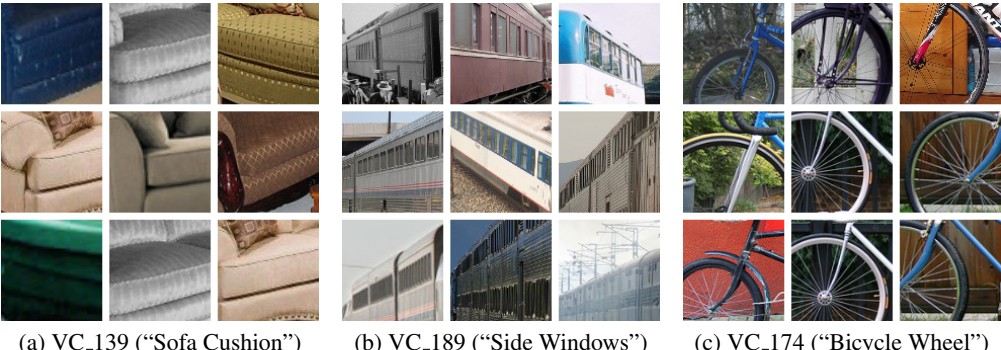

(a) VC_139 ("Sofa Cushion")  (b) VC_189 ("Side Windows")  (c) VC_174 ("Bicycle Wheel")

Figure 1: Visualizations of VCs. Each group consists of patches from original images closest to a VC. In general, these patches roughly correspond to semantic parts of objects, e.g., the cushion of a sofa (a), the side windows of trains (b) and the wheels of bicycles (c). All VCs are referred to by their indices (e.g., VC_139). We stress that VCs are learned in an unsupervised manner and terms like"sofa cushion" are inferred by observing the closest image patches and are used to describe them informally.

objects are represented by binary codes of VCs). This suggests that we can use VCs to represent new objects in terms of parts hence enabling few-shot learning.

But it is not obvious that VCs, as described in (Wang et al., 2017), can be applied to few-shot learning. Firstly, these VCs were learned independently for each object category (e.g., for cars or for airplanes) using deep network features from CNNs which had already been trained on data which included these categories. Secondly, the VCs were learned using large numbers of examples of the object category, ranging from hundreds to thousands. By contrast, for few-shot learning we have to learn the VCs from a much smaller number of examples (by an order of magnitude or more). Moreover, we can only use deep network features which are trained on datasets which do not include the new object categories which we hope to learn. This means that although we will extract VCs using very similar algorithms to those in (Wang et al., 2015) our motivation and problem domain are very different. To summarize, in this paper we use VCs to learn models of new object categories from existing models of other categories, while (Wang et al., 2015) uses VCs to help understand CNNs and to perform unsupervised part detection.

In Section 3, we will review VCs in detail. Briefly speaking, VCs are extracted by clustering intermediate-level raw features of CNNs, e.g., features produced by the Pool-4 layer of VGG16 (Simonyan & Zisserman, 2015). Serving as the cluster centers in feature space, VCs divide intermediate-level deep network features into a discrete dictionary. We show that VCs can be learned in the few-shot learning setting and they have two desirable properties when used for image encoding, which we call *category sensitivity* and *spatial patterns*.

More specifically, we develop an approach to few-shot learning which is simple, interpretable, and flexible. We learn a dictionary of VCs as described above which enables us to represent novel objects by their VC-Encoding. Then we propose two intuitive models: (i) nearest neighbor and (ii) a factorizable likelihood model based on the VC-Encoding. The nearest neighbor model uses a similarity measure to capture the difference between two VC-Encodings. The factorizable likelihood model learns a likelihood function of the VC-Encoding which, by assuming spatial independence, can be learned form a few examples. We emphasize that both these models are very flexible, in the sense that they can be applied directly to any few-shot learning scenarios. This differs from other approaches which are trained specifically for scenarios such as 5-way 5-shot (where there are 5 new object categories with 5 examples of each). This flexibility is attractive for real world applications where the numbers of new object categories, and the number of examples of each category, will be variable. Despite their simplicity, these models achieve comparable results to the state-of-the-art few-shot learning methods (using only the simplest versions of our approach), such as *learning a metric* and *learning to learn*. From a deeper perspective, our results show that CNNs have the

potential for few-shot learning on novel categories but to achieve this potential requires studying the internal structures of CNNs to re-express them in simpler and more interpretable terms.

Overall, our major contributions are two-fold:

(1) We show that VCs can be learned in the few-shot setting using CNNs trained on other object categories. By encoding images using VCs, we observe two desirable properties, i.e., category sensitivity and spatial patterns.

(2) Based on these properties, we present two simple, interpretable, and flexible models for few-shot learning. These models yield competitive results compared to the state-of-the-art methods on specific few-shot learning tasks and can also be applied directly, without additional training, to other few-shot scenarios.

## 2    RELATED WORK

Our work lies at the intersection of attempts to understand the internal representations of neural networks and research on few-shot learning. Therefore, we review here the previous literature on these two topics.

### 2.1    NEURAL NETWORK INTERNAL REPRESENTATIONS

Recently, there have been numerous studies aimed at understanding the behavior of neural networks and, in particular, to uncover the internal representations within CNNs. Some researchers try to visualize internal representations by sampling (Zeiler & Fergus, 2014), generating (Simonyan et al., 2013; Nguyen et al., 2016) or by backpropagating (Mahendran & Vedaldi, 2015) images in order to maximize the activations of the hidden units. A particularly relevant work by Zhou et al. (2015) shows that object and object parts detectors emerge in CNNs. Conversely, other works investigate the discriminative power of the hidden features of CNNs by assessing them on specific problems (Sharif Razavian et al., 2014; Bau et al., 2017; Agrawal et al., 2014; Yosinski et al., 2014). The overall findings suggest that deep networks have internal representations of object parts. The most relevant work to the paper is the study of VCs which discovered mid-level visual cues in the internal features of CNNs and showed relationships between these visual cues and semantic parts (Wang et al., 2015; 2017). This work is described in detail in Section 3

### 2.2    FEW-SHOT LEARNING

There have been growing attempts to perform few-shot learning motivated by attempts to mimic human abilities and to avoid some of the limitations of conventional data-demanding learning. An early attempt was made building on probabilistic program induction (Lake et al., 2015) and another attempt exploited object parts (Wong et al., 2017). More recent efforts at few-shot learning can be broadly categorized into two classes. The first is to design methods to embed the inputs into a feature space friendly to few-shot settings (Koch et al., 2015; Vinyals et al., 2016). Their goal is to find a good similarity measure (e.g., using Siamese networks) that can be applied rapidly to novel categories. The second is meta-learning which efficiently trains an ordinary model with the budget of few examples (Ravi & Larochelle, 2017; Finn et al., 2017). An alternative approach by Qiao et al. (2017) performs few-shot learning by estimating parameters of the prediction layer using regression from previously learned objects. We emphasize that the approach in our paper differs from these works, most of which are tailored for a few specific few-shot learning scenarios (i.e., test and train conditions must match), while our methods are simple and flexible, so that they work both in normal and almost all few-shot settings.

## 3    BACKGROUND: VISUAL CONCEPTS

In Wang et al. (2015), VCs were discovered as internal representations within deep networks which roughly correspond to mid-level semantic visual cues. These VCs play a core role in our work on understanding properties of CNNs and developing our interpretable few-shot learning models. In this section, we review how VCs were defined and learned in Wang et al. (2015). We will describe later in Section 4.1 how we modify VCs for few-shot learning.

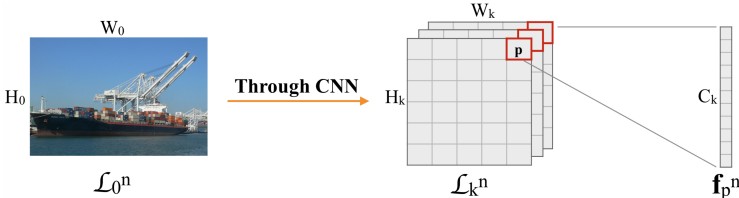

Figure 2: Key terms in the VC formalization. On the left is the $n$-th input image, defined on image lattice $\mathcal{L}_0^n$, with height $H_0$ and width $W_0$. In the middle is the lattice at the $k$th layer of the CNN for the $n$-th image, noted by $\mathcal{L}_k^n$, with height $H_k$ and width $W_k$. On the right is a feature vector at position $p$ in $\mathcal{L}_k^n$, noted by $\mathbf{f}_p^n$, with dimensionality $C_k$.

We first summarize the formalization of VCs, which are illustrated in Figure 2. CNNs contain a hierarchy of lattices $\mathcal{L}_l$, where $l \in \{0, 1, \dots\}$ stands for the layer of the lattice. In particular, the input image is defined over the lattice $\mathcal{L}_0$ and the lattice on which we derive VCs is specified by $\mathcal{L}_k$. We denote the spatial mappings from $\mathcal{L}_0$ to $\mathcal{L}_k$ by $\pi_{0 \mapsto k}$ and from $\mathcal{L}_k$ to $\mathcal{L}_0$ by $\pi_{k \mapsto 0}$. Then we define $\mathcal{F}_k^n = \{\mathbf{f}_p^n : p \in \mathcal{L}_k^n\}$ as the feature vector set at $\mathcal{L}_k^n$ of the $n$-th image where $p$ refers to a 2D position in the lattice $\mathcal{L}_k^n$. These feature vectors are computed by $\mathbf{f}_p^n = f(\mathbf{I}_{A(p')}^n)$, where the function $f$ is specified by the neural network and $\mathbf{I}_{A(p')}^n$ is a subregion of the $n$-th input image $\mathbf{I}^n$, centered on a point $p' = \pi_{k \mapsto 0}(p)$ on $\mathcal{L}_0^n$. In other words, the responses of all the channels in position $p$ constitute the feature vector $\mathbf{f}_p^n$. Then we have $\mathcal{F}_k = \cup_n \mathcal{F}_k^n$ from all images of interest. Note that by collecting feature vectors into $\mathcal{F}_k$, all spatial and image identity information is removed. Since layer $k$ is usually pre-selected for different network architectures for VCs applications (e.g., Wang et al. (2015) typically studied layer $k = 4$ for VGG16 net in their work), the subscript will be omitted in the remainder of the paper for simplicity.

Now we describe how VCs are extracted. The approach assumes that the VCs are represented by a population code of the CNN feature vectors. They are extracted using an unsupervised clustering algorithm. Since we first normalize the feature vectors into unit length, instead of using K-means as proposed in Wang et al. (2015), we assume that the feature vectors are generated by a mixture of von Mises-Fisher distributions (vMFM) (Hasnat et al., 2017) and learn this mixture by the EM algorithm (Banerjee et al., 2005). The goal is to maximize the likelihood function

$$P(\mathcal{F}|\alpha, \mu, \kappa) = \Pi_{m=1}^M \Sigma_{v=1}^V \alpha_{m,v} V_d(\mathbf{f}_m|\mu_v, \kappa_v), \tag{1}$$

where $M$ is the total feature vector count (i.e., the number of all feature vectors collected from all images of interest) and $V$ is the predefined VC (cluster) number. $V_d(\cdot)$ is the density function of the vMF distribution. $\mathbf{f}_m$ denotes each feature vector we get from the intermediate layer of a CNN (without image identity or spatial information). $\alpha$, $\mu$, and $\kappa$ are vMFM parameters and represent the mixing proportion, mean direction, and concentration values respectively.

We define the set of VCs by $\mathcal{V} = \{\mu_v : v = 1, \dots, V\}$ (i.e., by the mean directions of the learned vMFM distribution). Alternatively, since the $\{\mu_v\}$ have the same dimensionality as the $\{\mathbf{f}_m\}$, we denote a specific VC center by $\mathbf{f}_v = \mu_v$.

To help understand the VCs, we compute the cosine distances from the original feature vectors to the VCs as follows:

$$d_{p,v}^n = 1 - \frac{\mathbf{f}_p^n \cdot \mathbf{f}_v}{\left\|\mathbf{f}_p^n\right\|_2 \left\|\mathbf{f}_v\right\|_2} \tag{2}$$

where $d_{p,v}^n$ denotes the distance between feature vector $\mathbf{f}_p^n$ and the VC $v$ in the $n$-th image at position $p$, and we call them **VC distances**. We select those feature vectors with the smallest distances to each VC and trace them back to the original input image using $\pi_{k \mapsto 0}$. This yields "visualization patches" of VCs, shown in Figure 1. We observe that these patches roughly correspond to the semantic parts of objects, which justifies our assertion that VCs are semantic visual cues.

In the previous studies of VCs (Wang et al., 2015; 2017), the CNNs that were used to generate feature vectors were trained for a large scale object classification task that included the object categories of interest. Moreover, they extracted VCs using hundreds of images within a specific category of object, which resulted to category specific visual cues that were useful for interpreting CNN behaviors and

building novel models for semantic part detections. In more recent work (in preparation) VCs were used to encode semantic parts and objects using *VC-Encoding* that could be applied to detection tasks in the presence of occlusion. VC-Encoding is described in the next section. We emphasize that none of this prior work on VCs addressed few-shot learning and, by contrast, only addressed situations where there were many training examples.

# 4 FEW-SHOT LEARNING FROM VCS

This section describes the technical ideas of our paper. In Section 4.1, we introduce how we learn VCs in the few-shot setting. In Section 4.2, we introduce VC-Encoding and show its two desirable properties for few-shot classification tasks. Then in Section 4.3 and Section 4.4, we propose two simple, interpretable models for few-shot learning based on VC-Encoding.

## 4.1 FEW-SHOT VCS

It is not obvious that VCs can be applied to few-shot learning tasks where only few examples are available for each novel category. It is not possible to train the CNNs on the objects (as was done in Wang et al. (2015)) and also there may not be enough data to get good VC clusters. Hence we modify the way VCs are learned: we learn VCs from small number of examples of novel object categories using features from CNNs trained on other object categories. This is similar to how metric-learning and meta-learning are trained on large datasets which do not include the novel categories. This ensures that the CNN used for feature extraction has never seen the categories on which we will perform few-shot classification.

To extract VCs for the novel categories which only have few examples each, we pool feature vectors from the different categories together and perform the clustering algorithm on all of them. This gives us a little more data and encourages VC sharing between different categories. This improves data efficiency and also makes it easier to apply our VC models to multiple novel categories.

By the two modifications described above, we obtain **few-shot VCs**, i.e., VCs that are suitable for few-shot learning. This is critical for our application and differentiates this work from previous studies of VCs. Surprisingly, we find that we only need a few images (e.g., five images per category) to extract high quality VCs (see visualizations in Figure 3a) which, when used for VC-Encoding, possess similar desirable properties as the traditional VCs and hence are suitable for few-shot object classification task.

## 4.2 VC-ENCODING

We assume that objects can be decomposed into semantic parts. From the perspective of VCs, this means that most $\{\mathbf{f}_p\}$ should be assigned to *a single VC*. This requires specifying an explicit relationships between the $\{\mathbf{f}_p\}$ and the VCs. A natural choice is to compute the distances $d_{p,v}$ between the $\mathbf{f}_p$ and the $v$-th VC and threshold it to produce a binary value $b_{p,v}$ (i.e., $b_{p,v} = 1$ if $d_{p,v} < T$). We refer to $\mathcal{B} = \{b_{p,v} : p \in \mathcal{L}, v = 1, \ldots, V\}$ as the **VC-Encoding**. Note the image index $n$ is omitted here since the operations are identical for all images of interest. We use two criteria to specify a good encoding, *coverage* and *firerate*, defined as following:

$$coverage = \frac{\sum_{p \in \mathcal{L}} \max_v b_{p,v}}{|\mathcal{L}|} \tag{3}$$

$$firerate = \frac{\sum_{p \in \mathcal{L}} \sum_v b_{p,v}}{|\mathcal{L}|} \tag{4}$$

The choice of the encoding threshold $T$ is a trade-off between requiring sufficient coverage and a firing rate that is close to one. In practice, we choose $T$ for each testing trial by a grid-search with step size 0.001 which outputs the smallest threshold ensuring that the average $coverage >= 0.8$ for all few-shot training images. This yields the final VC-Encoding $\mathcal{B}$ used in our models, with the following desirable properties:

**Category Sensitivity** Despite the fact that the VCs are learned from a mix of images with different category labels, the first insight is that many VCs tend to fire ($b_{\cdot,v} = 1$) intensively for one or a

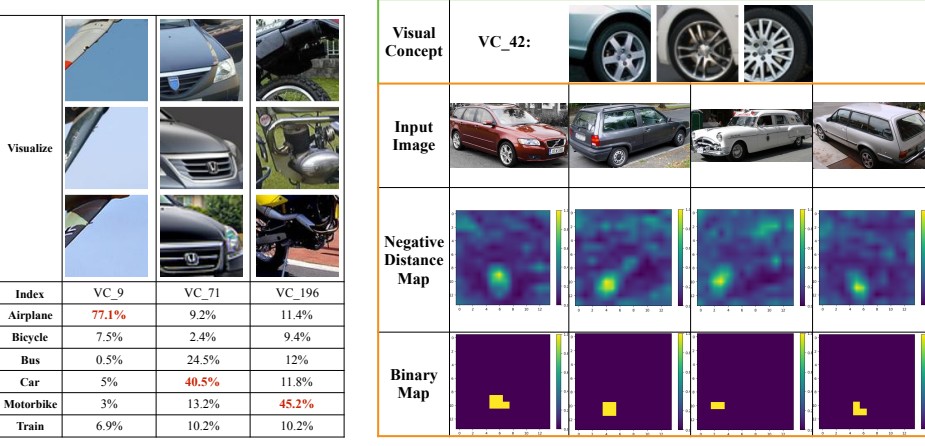

| Index | VC_9 | VC_71 | VC_196 |
|---|---|---|---|
| Airplane | **77.1%** | 9.2% | 11.4% |
| Bicycle | 7.5% | 2.4% | 9.4% |
| Bus | 0.5% | 24.5% | 12% |
| Car | 5% | **40.5%** | 11.8% |
| Motorbike | 3% | 13.2% | **45.2%** |
| Train | 6.9% | 10.2% | 10.2% |

(a) Category Sensitivity        (b) Spatial Patterns

Figure 3: Properties of VCs. In (a), we illustrate three VCs by their closest patches and their occurrence distributions over 6 object categories out of the 12 in PASCAL3D+ showing category sensitivity of VC-Encoding. In (b), we visualize the closest patches to VC_42 in the top green box. In the bottom orange box, we randomly select 4 images of cars and plot their negative distance maps and binary maps with respect to VC_42. More precisely, the negative distance map is given by $-d_{p,42}$ and is scaled to $(0, 1)$. The binary map is drawn based on $b_{p,42}$. See Section 4.2 for more details.

few specific object categories. In Figure 3a, we calculate the occurrence distributions of several VCs for 6 object categories out of the 12 in PASCAL3D+ (Xiang et al., 2014). In each column that represents a specific VC, the occurrence frequencies tend to be high for one or two object categories and low for the others. This suggests that VC identities can provide useful informations for object classification. Moreover, the corresponding visualized patches on the top of Figure 3a support our understanding that VCs have this category sensitivity because they capture the semantic parts that are specific for object categories.

**Spatial Patterns** The spatial pattern of VC firings is also indicative of the object category. Although spatial information is ignored during feature clustering, the learned VCs give binary maps that contain regular spatial patterns for images of the same category with relatively similar viewpoints (as shown in Figure 3b). This is consistent with the more general conjecture that the spatial patterns of semantic parts play a vital role in object recognition, and shows again that the VC-Encoding can capture the spatial patterns of the semantic parts to a certain extend.

Next, we build two simple few-shot learning models based on VC-Encoding learned from few examples.

### 4.3 NEAREST NEIGHBOR ON VC-ENCODINGS

First, we propose a simple template matching method which is similar to traditional nearest neighbor algorithms. The only novelty is that we use a similarity metric between VC-Encodings which is spatially "fuzzy" so that it can tolerate small spatial shifts of the semantic parts in images. Formally, the similarity metric takes the following form:

$$K(b, b') = \frac{1}{2}\left(\frac{\sum_{p,v} b_{p,v} \max_{q,q \in n(p)} b'_{q,v}}{\sum_{p,v} b_{p,v}} + \frac{\sum_{p,v} b'_{p,v} \max_{q,q \in n(p)} b_{q,v}}{\sum_{p,v} b'_{p,v}}\right) \quad (5)$$

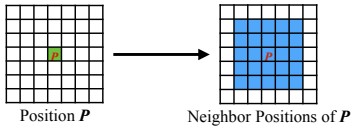

Position **P**      Neighbor Positions of **P**

where $K(b, b')$ is the similarity between the binary VC-Encodings $b$ and $b'$. $n(p)$ defines the set of neighboring positions of $p$ (as shown in Figure 4). During testing, we classify an image to the category of the training example with the largest similarity.

Figure 4: The green grid on the left is $p$. The blue grids on the right are $n(p)$.

One motivation for this method is that sequential convolutional operations carried in a neural network can be considered as embedding input images into a hierarchy of feature spaces. Each convolutional layer can be treated as a different level of decomposition of the inputs since the convolution along with non-linear activation, which take the form $\mathbf{Y} = \sigma(\mathbf{W} \cdot \mathbf{X} + \mathbf{B})$, is composed of matching templates $\mathbf{W}$ and using non-linear function $\sigma$ to filter out patterns based on the threshold $\mathbf{B}$. In light of this interpretation, and the two properties described earlier, it is reasonable that VC-Encoding will yield an explicit semantic decomposition.

### 4.4 Factorizable Likelihood Model

Apart from the intuitive nearest neighbor method, we present a second method which models the likelihood of the VC-Encoding. We observe that we can specify a distribution over the VC-Encoding $b_{p,v}$ using a bernoulli distribution with probability $\theta_{p,v}$. Following Naïve Bayes, we assume all the elements of the VC-Encoding $b$ are independent (making it possible to learn the distribution from a very small number of examples). Hence we can express the likelihood of $b$ as following:

$$\mathcal{L}(b|\theta) = \prod_{p,v} b_{p,v} \cdot \theta_{p,v} + (1 - b_{p,v}) \cdot (1 - \theta_{p,v}) \tag{6}$$

For each object category $y$, we derive a probabilistic distribution $\theta_y$ from the training examples. Thus the prediction of object category given the VC-Encoding $b$ is given by:

$$y_b = \max_y \mathcal{L}(b|\theta_y) \tag{7}$$

Note that by doing this, we are in fact implementing a discriminative model obtained from a generative distribution. We smooth each distribution $\theta_y$ using a Gaussian filter to guard against unlikely events.

## 5 Evaluations under Few-Shot Settings

Few-Shot learning is a very challenging task where humans perform much better than current algorithms. It requires the ability, or efficiency, to learn generalizable knowledge from strictly limited examples, such as a few training images.

Nevertheless, Section 4.1 suggests that a few images may be enough for learning object models when represented by VC-Encodings. Indeed our experiments show that both our two VCs-based few-shot learning models are competitive in performance with existing methods designed specifically for few-shot learning such as Ravi & Larochelle (2017). In addition, while previous few-shot methods are trained to work in specific few-shot scenarios, such as 5-way classifications, our methods can be applied to a large range of few-shot scenarios without additional training. The experimental results show that trained CNNs have the potential to recognize novel objects from few examples by exploiting VC-Encoding.

### 5.1 Mini-ImageNet

To assess the capability of our few-shot methods, we first evaluate them on a common few-shot learning benchmark, namely Mini-ImageNet. The Mini-ImageNet dataset was first proposed by Vinyals et al. (2016) as a benchmark for evaluating few-shot learning methods. It selects 100 categories out of 1000 categories in ImageNet with 600 examples per category. We use the split proposed by Ravi & Larochelle (2017) consisting of 64 training categories, 16 validation categories and 20 testing categories. In accordance with the convention for Mini-ImageNet, we perform numerous trials of few-shot learning during testing. In each trial, we randomly sample 5 unseen categories from a preserved testing set. Each category is composed of 5 training images for the 5-shot setting and 1 training image for the 1-shot setting. During evaluation, we randomly select 15 images for each category following Ravi & Larochelle (2017).

As Table 1 illustrates, we compare our methods against two baselines in line with the ones in Ravi & Larochelle (2017). In addition, we present the performances of state-of-the-art few-shot learning

| Method | 5-category | | 10-category |
| --- | --- | --- | --- |
| | 1-shot | 5-shot | 5-shot |
| **Baseline-finetune** | $28.86 \pm 0.54\%$ | $49.79 \pm 0.79\%$ | — |
| **Baseline-nearest-neighbor** | $41.08 \pm 0.70\%$ | $51.04 \pm 0.65\%$ | $39.89 \pm 0.48\%$ |
| **Pool3-nearest-neighbor** | $43.38 \pm 0.81\%$ | $55.33 \pm 0.75\%$ | $\mathbf{40.36 \pm 0.59\%}$ |
| **Matching Network** | $43.56 \pm 0.84\%$ | $55.31 \pm 0.73\%$ | — |
| **Meta-Learner LSTM** | $43.44 \pm 0.77\%$ | $60.60 \pm 0.71\%$ | — |
| **MAML** | $\mathbf{48.70 \pm 1.84\%}$ | $\mathbf{63.11 \pm 0.92\%}$ | — |
| **VC-nearest-neighbor (Ours)** | $\mathbf{46.39 \pm 1.09\%}$ | $58.84 \pm 1.12\%$ | $42.42 \pm 0.62\%$ |
| **VC-likelihood (Ours)** | $45.61 \pm 1.14\%$ | $\mathbf{63.07 \pm 1.02\%}$ | $\mathbf{45.11 \pm 0.66\%}$ |

Table 1: Average classification accuracies on Mini-ImageNet with $95\%$ confidence intervals. Evaluations of Baseline-finetune and Baseline-nearest-neighbor are from Ravi & Larochelle (2017). Pool3-nearest-neighbor stands for a nearest neighbor method based on raw Pool-3 features from the same VGG-13 as our methods. At the bottom are our factorizable likelihood method and nearest neighbor method based on VCs. Marked in bold at the top are the best published results for each scenario. Marked in bold at the bottom are our best results for the corresponding set-up. At the right is an extended setting for variance in the number of categories. Note in the last column we use the same models as in the middle column and we omit those that cannot be directly applied to this setting. We adopt the results for Matching Network from Ravi & Larochelle (2017).

methods including matching network (Vinyals et al., 2016), Meta-Learner (Ravi & Larochelle, 2017) and MAML (Finn et al., 2017). Regarding our methods, we train a VGG-13 on the training and validation set. The network is trained with the objective of cross entropy. We preserve $10\%$ images in each category to validate our network. We extract 200 VCs from the Pool-3 layer. The reason for choosing Pool-3 features is that a grid in Pool-3 lattices $\mathcal{L}_3$ correspond to a $36 \times 36$ patch in the original $84 \times 84$ image, which is a plausible size for a semantic part. For the gaussian filter used to smooth the factorizable likelihood model, we use a $\sigma$ of $1.2$. To directly examine the impact of the VCs, we also include the result of nearest neighbor matching using raw features from the pool-3 layer (referred as Pool3-Nearest-Neighbor in Table 1). Moreover, we attempt to evaluate the few-shot learning ability with the variance in the number of categories (as shown in the last column of Table 1) and further extend the few-shot learning evaluation with other randomly selected settings (as shown in Table 2), where we need to use exactly the same model as the middle columns in Table 1 trained once on the training and validation set.

The results show that our VCs-based methods compare well with current methods which were specifically designed for few-shot learning. If we contrast with meta-learning-based methods, we achieve higher accuracy than Meta-Learner both in the 1-shot and the 5-shot set-ups, while just slightly behind MAML. Compared with metric-based methods, which are more similar to ours, we marginally outperform the matching network, which is the state-of-the-art method of this category. These results confirm our assumption that low level visual cues within trained CNNs can naturally perform few-shot learning. We observe that on the 5-shot scenario, our likelihood model is significantly better than our nearest neighbor model. A possible explanation is that the likelihood model

| Method | 6-category | 8-category | 12-category |
| --- | --- | --- | --- |
| | 3-shot | 4-shot | 6-shot |
| **Baseline-nearest-neighbor** | $\mathbf{46.70 \pm 0.84\%}$ | $42.48 \pm 0.74\%$ | $38.49 \pm 0.49\%$ |
| **Pool3-nearest-neighbor** | $44.25 \pm 0.73\%$ | $\mathbf{43.30 \pm 0.73\%}$ | $\mathbf{38.77 \pm 0.53\%}$ |
| **VC-nearest-neighbor (Ours)** | $50.42 \pm 0.97\%$ | $46.39 \pm 0.74\%$ | $40.78 \pm 0.54\%$ |
| **VC-likelihood (Ours)** | $\mathbf{52.41 \pm 0.93\%}$ | $\mathbf{47.37 \pm 0.74\%}$ | $\mathbf{43.42 \pm 0.54\%}$ |

Table 2: Average classification accuracies on Mini-ImageNet with $95\%$ confidence intervals under randomly selected few-shot settings. All models used here and in Table 1 are the same set of models trained only once on the training set. Like the last column of Table 1, we omit those models which cannot be directly applied to various few-shot settings.

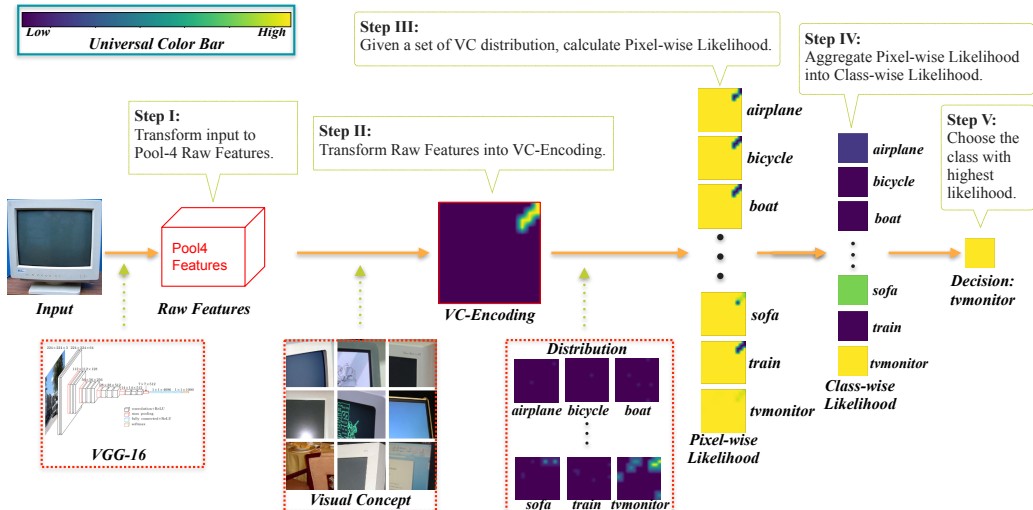

Figure 5: Visualizing the inference procedure of the factorizable likelihood model (using 1 VC for example). The original image is processed by part of VGG-16, and represented by its VC-Encoding. After calculating the likelihood for each pixel using distributions derived from a few examples, we obtain the pixel-wise likelihood. Then we use the likelihood to make the final decision. For better visualization, we rescale the variance of aggregated likelihood to 1. For all the visualizations, we use the same Universal Color Bar. This figure is best viewed in color.

combines several training examples into a distribution while nearest neighbor can only use examples individually. For instance, if a front wheel of cars appears in one training example and a rare wheel occurs in another example, the likelihood model can combine these two wheels into a distribution while nearest neighbor can only match testing examples with either the front wheel or the rare wheel. Finally, using the same model in the middle columns of Table 1, some previous methods like Meta-Learner LSTM are not applicable to various extended settings in Table 2 and the last column in Table 1. In fact, these methods can directly deal with only changes in the number of shots but cannot deal easily with changes in the number of categories. But unlike these methods, our few-shot learning methods based on VC-Encoding can be easily extended (with minimal re-training) to any number of shots and any number of categories.

## 5.2    PASCAL3D+

To delve deeper into our methods, we apply them to PASCAL3D+, a dataset with larger high quality images than Mini-ImageNet. PASCAL3D+ (Xiang et al., 2014) is a dataset augmenting 12 rigid categories of the PASCAL VOC 2012 (Everingham et al.). It was originally tailored for 3D object detection and pose estimation. We choose PASCAL3D+ as our testbed since it provides high quality images with comparable image size to ImageNet. We interpret our few-shot recognition mainly by visualizing every step of the inference. With input images of sufficient sizes, we can obtain large VC-Encoding distribution maps whose visualizations are easy for humans to interpret.

The simplicity of our methods makes the inference process of few-shot recognition very transparent. On PASCAL3D+, we first qualitatively analyze this procedure based on VCs. In Figure 5, we visualize every step of our method based on an example VC. Among the closest patches, the corner of the TV Monitor occurs most frequently. So we assume this VC relate to the corner of TV Monitor. The distributions of this VC suggest it mainly responds to the upper right of TV Monitors since only the upper right corner of TV Monitor's distribution map shows high frequency (see the distributions in Figure 5). Using this VC, we convert the original deep network features into VC-Encoding. The VC-Encoding implies this VC fires on the upper right corner of the input (see the VC-Encoding map in Figure 5). After calculating the pixel-wise likelihood using the distributions from a few images, it is clear that except for the TV Monitor, each category has low likelihood in the area of the corner

| Method | Number of VCs | 12-category | |
| --- | --- | --- | --- |
| | | 1-shot | 5-shot |
| **Pool4-nearest-neighbor** | – | **36.12%** | 52.30% |
| **Pool4-SVM** | – | 32.66% | **52.46%** |
| **VC-likelihood (Ours)** | 120 | 39.25% | 64.37% |
| **VC-likelihood (Ours)** | 200 | **40.02%** | 66.00% |
| **VC-likelihood (Ours)** | 300 | 39.23% | **66.47%** |
| **VC-nearest-neighbor (Ours)** | 120 | 40.74% | 58.52% |
| **VC-nearest-neighbor (Ours)** | 200 | **42.36%** | 59.47% |
| **VC-nearest-neighbor (Ours)** | 300 | 41.18% | **61.07%** |

Table 3: Average classification accuracies on PASCAL3D+. At the top is the group of baseline methods including nearest neighbor and Exemplar-SVM based on Pool-4 features from the same VGG-16 used in our methods. In the middle are our factorizable likelihood models using different number of VCs. At the bottom are our VCs-based nearest neighbor models. Marked in bold are the best results within each group for each scenario.

(dark parts of pixel-wise likelihood maps in Figure 5). Finally, we aggregate the likelihood and make the correct classification decision.

Meanwhile, we quantitatively evaluate our methods on PASCAL3D+. More specifically, we employ PASCAL3D+ as our testing dataset. For training, we use the ImageNet (Deng et al., 2009) classification dataset without object categories related to 12 rigid categories (956 categories left). We train an ordinary VGG-16 as our starting point which achieves $71.27\%$ top-1 accuracy. For testing, we crop the objects out using annotated bounding boxes provided by Xiang et al. (2014) and resize them into $224 \times 224$. Then we use Pool-4 features produced by the VGG-16 to implement our few-shot methods instead of the Pool-3 features used in 5.1. The main reason for this change is the increased input image size of 224 in PASCAL3D+, which suggests that Pool-4 features will be better for capturing the semantic parts. As a comparison, we propose 2 baseline models. One (referred to as Baseline-nearest-neighbor in Table 3) is a nearest neighbor method based on raw Pool-4 features using the cosine distance metric. The other (referred to as Baseline-SVM in Table 3) is an Exemplar-SVM trained using hinge loss. Both of these baselines use the same pre-trained VGG-16 as our methods. During evaluation, we set 20 trials of both 5-shot and 1-shot learning over 12 categories on PASCAL3D+. We also assess our methods using different numbers of VCs to see impacts of the number of VCs. The results are shown in Table 3.

In light of our testing results, we conclude that VC-Encoding is a useful semantic decomposition of images into parts. In general, our methods based on VCs significantly outperform two baselines. In particular, the difference between our nearest neighbor methods and the baseline nearest neighbor methods is due to the use of VCs (e.g., by first transferring the raw feature vectors into VC distances, and by thresholding to get the VC-Encoding). Thus, we claim that decomposing fuzzy features (i.e., deep network features) into explicit semantic cues (i.e., the VCs) improves both interpretability and performance. We also notice that our methods are not sensitive to the number of VCs since changes of the number of VCs only cause slight differences among accuracies.

## 6 CONCLUSION

In this paper we address the challenge of developing simple interpretable models for few-shot learning exploiting the internal representations of CNNs. We are motivated by VCs Wang et al. (2015) which enable us to represent objects in terms of VC-Encodings. We show that VCs can be adapted to the few-shot learning setting where the VCs are extracted from a small set of images of novel object categories using features from CNNs trained on other object categories. We observe two properties of VC-Encoding, namely category sensitivity and spatial pattern, which leads us to propose two novel, but closely related, methods for few-shot learning which are simple, interpretable, and flexible. Our methods show comparable performances to the current state-of-the-art methods which are specialized for specific few-shot learning scenarios. We demonstrate the flexibility of our two models by showing that they can be applied to a range of different few-shot scenarios with

minimal re-training. In summary, we show that VCs and VC-Encodings enable ordinary CNNs to perform few-shot learning. We emphasize that in this paper we have concentrated on developing the core ideas of our two few-shot learning models and that we have not explored variants of our ideas which could lead to better performance by exploiting standard performance enhancing tricks, or by specializing to specific few-shot challenges. Future work includes improving the quality of the extracted VCs and extending our approach to few-shot detection.

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

# A    ABLATION STUDIES FOR VC-BASED METHODS

Although they were not tailored for specific few-shot learning scenarios, our VC-Based methods have comparable performance to previous methods while being more flexible and interpretable. Since we are proposing a new method for few-shot learning, we carried out ablation studies to understand the influences of different model components on the effectiveness of our approach.

**Encoding Against Distance.**    In Section 4.2, we transformed real-valued distances from VCs into binary *VC-Encoding* by a threshold, which was dynamically determined to satisfy our requirement for $coverage$ and $firerate$. We claim that this encoding is beneficial to learning with limited budgets, e.g., few shots. First, it enables us to implement the *Factorizable Likelihood Model*, which relies on binary encoding and shows better results than nearest neighbor methods. Second, by thresholding the distances, we are filtering out the noises of model estimations based on few training examples. This also helps the nearest neighbor model. In Table 1, we compare our models based on *VC-Encoding* with a model based on real-valued distances (referred to as VC-Distance in Table 1), which uses the cosine distances as the metric. The results show that while despite being better than the original CNN features (Pool3-nearest-neighbor in Table 1), the distance-based model is not as effective as binary coded models.

| Method | 5-category | |
|---|---|---|
| | 1-shot | 5-shot |
| Pool3-nearest-neighbor | $43.38 \pm 0.81\%$ | $55.33 \pm 0.75\%$ |
| VC-nearest-neighbor | $\mathbf{46.39 \pm 1.09\%}$ | $58.84 \pm 1.12\%$ |
| VC-likelihood | $45.61 \pm 1.14\%$ | $\mathbf{63.07 \pm 1.02\%}$ |
| VC-distance | $44.26 \pm 1.01\%$ | $56.90 \pm 1.02\%$ |

Table 1: VC models using distances and *VC-Encoding* on Mini-ImageNet. VC-distance stands for a nearest neighbor model using distances regarding VCs and the cosine distance metric. VC-nearest-neighbor and VC-likelihood are our proposed models based on *VC-Encoding*. Pool3-nearest-neighbor is a nearest neighbor model using original CNN features. Marked in bold are the best result for each few-shot set-up.

**Clustering Methods.**    VCs are obtained without supervision by clustering features extracted by CNNs. In our proposed models, we use the clustering method based on a mixture of von Mises-Fisher distribution. By learning this distribution through the EM algorithm, we can obtain cluster centers that remain on the unit sphere. This is mathematically reasonable since we expect the features to distribute on the unit sphere. In Table. 2, we compare the results using von Mises-Fisher distribution with those using K-Means clustering on the 5-Shot 5-category learning task. The results show that for both VC-Likelihood and VC-nearest-neighbor K-Means is just slightly behind the von Mises-Fisher distribution. Hence, we state that VC-based models are robust to different clustering methods.

| Clustering | Likelihood | Nearest Neighbor |
|---|---|---|
| K-Means | $62.51 \pm 0.89\%$ | $\mathbf{59.60 \pm 1.04\%}$ |
| von Mises-Fisher | $\mathbf{63.07 \pm 1.02\%}$ | $58.85 \pm 1.12\%$ |

Table 2: Different clustering for each model we proposed on the 5-shot 5-category setting on Mini-ImageNet. Likelihood refers to the *Factorizable Likelihood Model*. Marked in bold are the best result for each model.

**Scale of Semantic Parts.**    While our VCs are extracted without any supervision, we find that they are related to semantic parts of objects. However, semantic parts can have various scales in images. To keep the simplicity of our model, we choose a single scale of semantic parts. In terms of the CNN we used, we choose to use Pool-3 features of VGG-13. In Table. 3, we test different scales of VCs, i.e., features from different layers, under the 5-Shot 5-Category few-shot learning setting.

We see that our proposed scale is the most effective of the different scales. In addition, we notice that while the performance of the baseline method, which uses the original features for nearest neighbor, drops drastically as scale changes, the performances our VC-based models, especially our Likelihood model, decay more gradually. This also reflects the robustness of the VC-based models.

| Layer | Original | Nearest Neighbor | Likelihood |
|---|---|---|---|
| **Pool-3** | $55.33 \pm 0.75\%$ | $58.84 \pm 1.12\%$ | $63.07 \pm 1.02\%$ |
| **Pool-2** | $43.34 \pm 1.02\%$ | $46.92 \pm 0.93\%$ | $57.45 \pm 0.98\%$ |
| **Pool-1** | $33.60 \pm 0.82\%$ | $41.63 \pm 0.97\%$ | $53.62 \pm 0.94\%$ |

Table 3: Results of using feature from different layers of VGG-13 on Mini-ImageNet. Original refers the nearest neighbor model based on the original VGG features. Nearest Neighbor denotes the our nearest neighbor model based on *VC-Encoding*. Likelihood is our *Factorizable Likelihood Model*.

## B    STATISTICS FOR VC DISTRIBUTION

In this section we illustrate more VCs (in Figure 1) and their distributions over object categories (in Table 4).

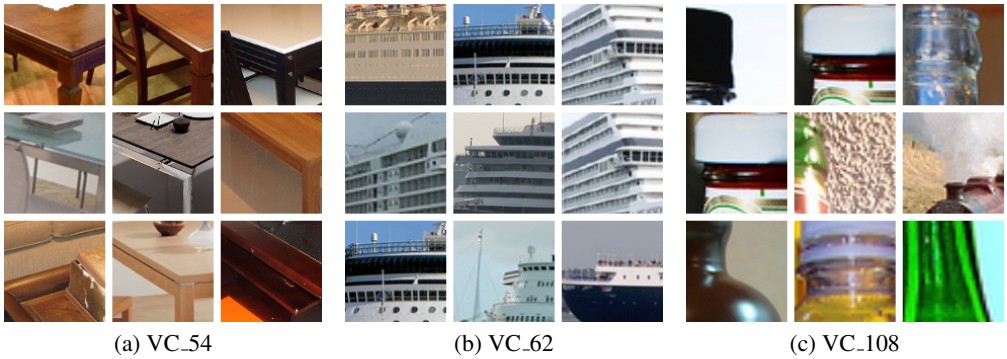

(a) VC_54                 (b) VC_62                 (c) VC_108

Figure 1: Visualizations of VCs. Each group consists of patches from original images closest to a VC. In general, these patches roughly correspond to semantic parts of objects, e.g., table corners (a), side windows of boats (b) and bottlenecks (c). All VCs are referred to by their indices.

| Index | Plane | Bike | Boat | Bottle | Bus | Car | Chair | Table | Motor | Sofa | Train | Monitor |
|---|---|---|---|---|---|---|---|---|---|---|---|---|
| VC_54 | 3.3 | 1.4 | 0.6 | 3.9 | 0.5 | 3.3 | 11.6 | **58.2** | 1.7 | 4.7 | 1.3 | 9.5 |
| VC_62 | 8.2 | 1.7 | **56.7** | 0.8 | 10.2 | 4.5 | 1.1 | 0.7 | 1.4 | 1.7 | 9.7 | 3.3 |
| VC_108 | 5.7 | 0.9 | 5.3 | **49.0** | 7.3 | 6.2 | 1.4 | 1.6 | 1.3 | 1.5 | 9.6 | 10.2 |

Table 4: Distribution of VCs over object categories. The object categories are from PASCAL3D+. Each row is the distribution of the VC over 12 categories (in percent). Marked in bold are the object categories with highest frequency for each VC. Plane is short for Aeroplane. Motor is short for Motorbike. Table is short for Diningtable. Monitor is short for TVMonitor. Bike stands for Bicycle.

