# OpenReview forum: "Unleashing the Potential of CNNs for Interpretable Few-Shot Learning"
_ICLR.cc/2018/Conference — Reject_

### Official Review · AnonReviewer2 · 2017-11-27
**Not enough novelty and important details unclear**

**Rating:** 5
**Confidence:** 4

**Review:**

My main concern for this paper is that the description of the Visual Concepts is completely unclear for me. At some point I thought I did understand it, but then the next equation didnt make sense anymore... If I understand correctly, f_p is a representation of *all images* of a specific layer *k* at/around pixel "p", (According to last line of page 3). That would make sense, given that then the dimensions of the vector f_p is a scalar (activation value) per image for that image, in layer k, around pixel p. Then f_v is one of the centroids (named VCs). However, this doesnt seem to be the case, given that it is impossible to construct VC activations for specific images from this definition. So, it should be something else, but it does not become clear, what this f_p is. This is crucial in order to follow / judge the rest of the paper. Still I give it a try.

Section 4.1 is the second most important section of the paper, where properties of VCs are discussed. It has a few shortcomings. First, iIt is unclear why coverage should be >=0.8 and firerate ~ 1, according to the motivation firerate should equal to coverage: that is each pixel f_p is assigned to a single VC centroid. Second, "VCs tent to occur for a specific class", that seems rather a bold statement from a 6 class, 3 VCs experiment, where the class sensitivity is in the order 40-77%. Also the second experiment, which shows the spatial clustering for the "car wheel" VC, is unclear, how is the name "car wheel" assigned to the VC? That has have to be named after the EM process, given that EM is unsupervised. Finally the cost effectiveness training (3c), how come that the same "car wheel" (as in 3b) is discovered by the EM clustering? Is that coincidence? Or is there some form of supervision involved?


Minor remarks
- Table 1: the reported results of the Matching Network are different from the results in the paper of Vinyals (2016).
- It is unclear what the influence of the smoothing is, and how the smoothing parameter is estimated / set.
- The VCs are introduced for few-shot classification, unclear how this is different from "previous few-shot methods" (sect 5).
- 36x36 patches have a plausible size within a 84x84 image, this is rather large, do semantic parts really cover 20% of the image?
- How are the networks trained, with what objective, how validated, which training images? What is the influence of the layer on the performance?
- Influence of the clustering method on VCs, eg k-means, gaussian, von-mises (the last one is proposed)?

On a personal note, I've difficulties with part of the writing. For example, the introduction is written rather "arrogant" (not completely the right word, sorry for that), with a sentence, like "we have only limited insights into why CNNs are effective" seems overkill for the main research body. The used Visual Concepts (VCs) were already introduced by other works (Wangt'15), and is not a novelty. Also the authors refer to another paper (about using VCs for detection) which is also under submission (somewhere). Finally, the introduction paragraph of Section 5 is rather bold, "resembles the learning process of human beings"? Not so sure that is true, and it is not supported by a reference (or an experiment).

In conclusion:
This paper presents a method for creating features from a (pre-trained) ConvNet.
It clusters features from a specific pooling layer, and then creates a binary assignment between per image extracted feature vectors and the cluster centroids. These are used in a 1-NN classifier and a (smoothed) Naive Bayes classifier. The results show promising results, yet lack exploration of the model, at least to draw conclusions like "we address the challenge of understanding the internal visual cues of CNNs". I believe this paper needs to focus on the working of the VCs for few-shot experiments, showing the influences of some of the choices (layer, network layout, smoothing, clustering, etc). Moreover, the introduction should be rewritten, and the the background section of VCs (Sect 3) should be clarified. Therefore, I rate the current manuscript as a reject.

After rebuttal:
The writing of the paper greatly improved, still missing insights (see comments below). Therefore I've upgraded my rating, and due to better understanding now, als my confidence.

---

> ### Author Response · Authors · 2018-01-05
> **Reply to AnonReviewer2**
>
> Thanks for your detailed comments!
>
> > Unclear description of VCs.
> R: We are sorry for the difficulties you encountered in reading this paper. We have thoroughly revised the whole paper and, in particular, improved the clarity of Section 3 and Section 4. For the specific issue you mentioned, L_k is the set of positions in the k-th layer of the CNN for an input image. That means an element p in this set is “a specific position” of the feature maps from “a specific image”. If we assume that our network has C_k feature channels at the k-th layer, then f_p will be a C_k-dimensional vector.
>
> > Problems on the properties of VCs
> R:
>     1. The validity of property 1: On the first property, we list the statistics of 6 categories in Figure. 3(a) for a concise illustration of this property (We have added more examples and stats on more VCs and object categories in the appendix of the updated paper). Our conclusions still hold for more categories. The occurrence percentage of the sensitive categories, though “only” in the order of 40-77%, substantially outnumbers the percentages of other categories and hence can provide useful information for classification. We also replaced the word “dominate” by“fire intensively” to make the meaning clearer.
>
>     2. Misunderstanding on the interpretation of VCs such as “car wheel”: On the second and the third property (removed in the updated version), there is a misunderstanding of the "car wheel" VC. The VCs are extracted in an unsupervised manner (e.g. no spatial and image identity information) and are indexed by an integer. We used the term "car wheel" to describe the VC after we visualized it and found the image patches correspond to car wheels. This term was only used informally to give an intuition for the semantic information the VCs represent. It is not a supervision used by the model. In the updated paper, we replaced the informal names with VC index in Figure 1 and Figure 3 to avoid further confusions.
>
> > Minor remarks
> R:
>     1. Please note that results in the paper of Vinyals (2016) used a private split of  Mini-ImageNet. So it's impossible to re-implement their settings. In this paper, we use a public split of Mini-ImageNet proposed by Ravi & Larochelle, 2017. The results for Matching Network are the same as Ravi & Larochelle, 2017 (We have added remarks for this in the caption of Table. 1).
>
>     2. In the original paper, we stated the parameter of our smoothing filter in the second paragraph of Section 5.1 ("For the Gaussian filter used to smooth the factorizable likelihood model, we use a \sigma of 1.2"). Also, we stated in the last sentence of Section 4.4 (original Section 4.3), the use of smoothing is to overcome the spareness of per pixel firing rate in our VC-likelihood model and help to improve the generalizability during testing. With this smoothing operation, our model can better handle small shiftings and deformations in the images. In our experiments, without the smoothing, our VC-likelihood model scores 61.84% in the 5-category 5-shot setting (see Table 1 for comparison). It is just slightly behind the model with smoothing. We use the smoothed results throughout the paper since the smoothing is also part of our model.

---

> > ### Author Response · Authors · 2018-01-05
> > **Reply to AnonReviewer2 (Continued)**
> >
> >     3. Previous methods are either learning a metric among categories or learning to learn a classifier within few examples. Our method is learning a composition of semantic concepts, which are extracted from few examples. We argue that our approach differs from previous one in terms of methodology. Also, our models possess the flexible (Shown in Table 2) and interpretable (shown in Figure 5) characters unlike alternative models.
> >
> >     4. We agree that semantic parts have various scales. However, for the simplicity of our method, we choose the layer with the most reasonable scale of VCs. In the appendix, we carried out experiments using features from different layers. We see from Appendix Table 3 that using features from the proposed Pool-3 layer achieves the best result. Meanwhile, we notice that compared with the original features, our VC-based methods are more robust to various scales.
> >
> >     5. We are sorry that we only described the details of network training for PASCAL3D+ in Section 5.2 of the original version. We have added more details in the description for Mini-ImageNet in Section 5.1. The network was trained on the training split of Mini-ImageNet with the objective of cross entropy. Specifically, we use the training split of the published Mini-ImageNet split from Ravi & Larochelle 2017. We preserve some images per category in the training split to validate our network.
> >
> >     6. VCs were proposed in Wang et al. (2015). We adopted the basic ideas of VCs from Wang et al. (2015) but adapted them to few-shot learning problems. In the appendix, we compared results using K-Means clustering and von Mises-Fisher clustering. While von Mises-Fisher clustering is mathematically more reasonable based on our assumptions, we find that there are minor differences between different clustering methods empirically. This shows that our VC-based models are robust to different clustering methods.
> >
> > > Problems on the writing
> > R: We have revised the writing of our paper thoroughly, removed many of the “bold” claims, and toned down the “arrogance”. Instead, we concentrated on giving a clearer description of our work, including clarifying the novelty and effectiveness of our models. Please refer to the update list to see the modifications.

---

### Official Review · AnonReviewer1 · 2017-11-27
**Incremental paper, but good results**

**Rating:** 7
**Confidence:** 4

**Review:**

The paper adds few operations after the pipeline for obtaining visual concepts from CNN as proposed by Wang et al. (2015). This latter paper showed how to extract from a CNN some clustered representations of the features of the internal layers of the network, working on a large training dataset. The clustered representations are the visual concepts. This paper shows that these representations can be used as exemplars by test images, in the same vein as bag of words used word exemplars to create the bag of words of unseen images.

 A simple nearest neighborhood and a likelihood model is built to assign a picture to an object class.

The results a are convincing, even if they are not state of the art in all the trials.
The paper is very easy to follows, and the results are explained in a very simple way.


Few comments:
The authors in the abstract should revise their claims, too strong with respect to a literature field which has done many advancements on the cnn interpretation (see all the literature of Andrea Vedaldi) and the literature on zero shot learning, transfer learning, domain adaptation and fine tuning in general.

---

> ### Author Response · Authors · 2018-01-05
> **Reply to AnonReviewer1**
>
> Thanks for your comments! We cited previous works on CNN internal representations in Section 2 in the original version, and we modified the paper to cite these works in the introduction as well. The revision will be reflected in our updated version.

---

### Official Review · AnonReviewer3 · 2017-11-30
**A paper with limited novelty**

**Rating:** 4
**Confidence:** 5

**Review:**

The paper proposes a method for few-shot learning using a new image representation called visual concept embedding. Visual concepts were introduced in Wang et al. 2015, which are clustering centers of feature vectors in a lattice of a CNN. For a given image, its visual concept embedding is computed by thresholding the distances between feature vectors in the lattice of the image to the visual concepts. Using the visual concept embedding, two simple methods are used for few-shot learning: a nearest neighbor method and a probabilistic model with Bernoulli distributions. Experiments are conducted on the Mini-ImageNet dataset and the PASCAL3D+ dataset for few-shot learning.

Positives:
- The three properties of visual concepts described in the paper are interesting.

Negatives:
- The novelty of the paper is limited. The idea of visual concept has been proposed in Wang et al. 2015. Using a embedding representation based on visual concepts is straightforward. The two baseline methods for few-shot learning provide limited insights in solving the few-shot learning problem.

- The paper uses a hard thresholding  in the visual concept embedding. It would be interesting to see the performance of other strategies in computing the embedding, such as directly using the distances without thresholding.

---

> ### Author Response · Authors · 2018-01-05
> **Reply to AnonReviewer3**
>
> Thanks for your comments!
>
> > Lack of novelty in this work.
> R: The novelties of this work lie both in new results for VCs and new methods for few-shot learning. Sorry that we did not make this clear enough in the original submission.
>
>     1. New results for VCs. It is not obvious that the original VCs described in Wang et al. 2015 can be applied to few-shot learning. We adapt VCs for few-shot learning and our key findings, which are critical for few-shot learning, were not addressed in Wang et al. 2015.
>
>         a. Extracting VCs from CNNs trained on different object categories: We learn VCs for objects from a CNN trained on a different set of objects categories (e.g., we learn VCs for vehicles with a CNN trained on a non-vehicle dataset). By contrast,  in Wang et al. 2015, the VCs were extracted from a subset of the object categories on which the CNN was trained.
>
>         b. Extracting VCs from few examples: In this work, we extract VCs from very few examples per category, but Wang et al. 2015 used orders of magnitude more (around 1000 versus 25). Surprisingly, we find that these “few-shot” VCs, when used for VC-encoding, possess similar desirable properties as the traditional VCs and hence are suitable for few-shot object classification task.
>
>         c. Extracting VCs without knowing the object category: In this work, we extract VCs without knowing the object category (e.g., by pooling feature vectors from different categories together and clustering over them). But in Wang et al. 2015, VCs were extracted separately for each object category to obtain category specific VCs. This modification provides sufficient samples to learn high-quality VCs in few-shot setting and encourages VC sharing among different categories. This is useful for improving data efficiency and also makes it easier to apply our VC models on multiple novel categories directly.
>
>         d. We found novel properties of VCs, i.e., category sensitivity and spatial pattern, which support the extended application of VCs. We describe these properties in detail in the second and the third paragraph of Section 4.2.
>
>     2. A new approach to few-shot learning. Unlike previous few-shot learning methods, we formulate few-shot learning in terms of compositions of semantic visual cues (i.e., parts). This differs from standard approaches like metric-based methods (e.g., matching network) and meta-learning based methods (e.g., MAML or Meta-Learner). Moreover, as stated in Table. 2, our proposed models are simple and very flexible (i.e., the same model can be applied with minimal changes to different problem settings, such as 5-category classification, 6-category classification, etc.). We argue that flexibility is an important characteristic of few-shot learning for real-world applications.
>
> > The hard threshold in the VC-Embedding.
> R: First, this threshold is for binary encoding of VCs which makes it possible to learn simple Bernoulli distributions  (i.e, we only need to learn a distribution over binary variables which needs very little data). As shown in our experiments, this model is slightly better than nearest neighbor. Second, we compared Nearest Neighbor using distances to VCs vs VC-Encoding. As shown in Appendix Table 1, the accuracy of using distances directly is slightly worse than using VC-Encoding, but is still better than using original features from CNN.

---

### Public Comment · (anonymous) · 2017-12-16
**A Difficult to Reproduce Paper**

After careful and thorough analysis of this paper, we believe this paper is difficult to reproduce for several reasons. First, the paper is vague at times while attempting to explain certain concepts. The definition of visual concepts is not clear, and could use more elaboration. Second, this paper is not very well self-contained, and required consulting the source paper to properly understand the idea of visual concepts. Other concepts, such as the data partition, are also not explained in detail and required consulting the source for more elaboration. These issues increase the difficulty of reproducing this paper.


Unfortunately, our group could not reproduce the results of this paper. One explanation is due to the lack of source code. Without the source code, it becomes much more difficult to output the same results. Thus our group had to implement the algorithm by hand. However, we noticed that the methodology contained few hiccups. For one thing, there was a lack of specificity within the paper at times that put the burden of choosing many non-obvious design choices on the us. For instance, many hyper parameters could use more explanation. One example occurs when the variable b’ is introduced in the function K(b, b’). It’s not very clear what b’ represents at first, and it took a bit of pondering to realize it is for another input VC-Encoding argument. Variables such as b’ could be provided with more detail so as to ease any attempts at reproducibility.

Another issue is that the method used to derive some values such as the threshold for creating the visual encodings is missing. Though a general process is described to find the threshold value, the exact value is not given. Because the threshold is dependent on other values, that are also not provided, it makes it difficult to estimate a value that would have the same results as this paper.  Finally, the formulas provided seem to be quite resource intensive, however hardware used or the amount of computing power required to train and use these models are not provided. It would be incredibly beneficial to know what their hardware specifications are, since our basic implementation of their similarity function ran with a running time that grew quadratically with the number of visual concepts, linearly with the number of pixels, and linearly with the number of neighbours for each pixel. This turned out to be prohibitively expensive to run off a simple laptop, taking over two hours to build and classify with only 15 training images and 15 validation images, and an implementation on  a GPU with 80 images took over 30 hours to complete. Maybe our implementation is poorly optimized? Unfortunately we cannot know unless we have hardware or computational details to compare to.

Overall while this paper shows potential for a extremely interesting new learning algorithm, the results are difficult to duplicate. Because of the ambiguity of some of the terms and concepts, it was challenging to follow the protocol and replicate the values that were proposed by the authors. For these reason, we feel this paper could be improved by providing more parameter values, and releasing the source code so that the results may be replicated.

---

> ### Author Response · Authors · 2017-12-21
> **Reply to the Reproducibility Issue**
>
> Thanks for your interests in our paper, and sorry to hear about the difficulty you’ve met in reproducing our results. We’d like to clarify your confusions about our proposed method.
>
> > Definition of VC & Not well self-contained.
> While we have cited Wang et al. 2015 to acknowledge the original idea of visual concepts, we offered a detailed review of this key background in Section 3. Please see the description and figures in that section for the in-detail explanation of VC. Moreover, we’d appreciate a lot if you can share more specific problems you have had in understanding VC to help us make the definition clearer.
>
> > Data Partition.
> Please note in the first paragraph of Section 5.1 we have stated that “we use the split proposed by Ravi & Larochelle (2017) consist of 64 training classes, 16 validation classes and 20 testing classes”, which is a broadly used public split in few-shot learning evaluations.
>
> > Source code
> We planned to open-source our code after the paper gets accepted, but since you raised the reproducibility issue, we will publish it soon. Nevertheless, we think this work is fairly easy to implement even from scratch. Most components of our model can be implemented using standard ML toolkit, e.g. clustering method can be found in the sklearn package(http://scikit-learn.org/) and spherecluster (https://github.com/clara-labs/spherecluster). After VC extraction, both the nearest neighbor and likelihood method can also be easily implemented by hand or using the standard toolkit.
>
> > Hiccups in Methodology
> Please note that in the first paragraph of Section 4.2 (right behind equation (4)), we have stated that “ K(b, b’) is the similarity between the binary VC-encodings b and b’ ”. That is to say, b’ is a VC-encoding which we have defined in Section 4.1. Other hyperparameters are also stated in Section 5.1. We’d appreciate it a lot if you can share other specific hiccups to help us revise the writing.
>
> > Method to Obtain the Threshold
> Please note that in the second paragraph of Section 4.1, we have stated that “the threshold will be found by a grid-search which outputs the smallest threshold ensuring that coverage >= 0.8 with step size 0.001”. This means the threshold is calculated in real-time for each trail. Thus we can’t provide you with an actual number of it. Also, we have defined coverage and firerate in the first paragraph of this section, which we guess would be the “other values” you have mentioned.
>
> > Hardware Specification and Computational Details
> In our experiment, we use 1 NVIDIA Titan X GPU. Regarding the computational cost, there are two core formulas in our model, equation (4) and (5), both have complexity  O(NV), where N is the number of pixels (h by w) and V is the number of visual concepts. In practice, the order of magnitude for both N and V is 2. If you vectorize all the operations, it will be quite fast. Thus, in our experiment, each trail of few-shot learning can be finished less than 2 minutes. It would be good if you can share your implementation so that we could help to troubleshoot your efficiency problem.
>
> Overall, we thank you for your valuable feedback. Further specific suggestions will also be highly appreciated.

---

### Author Response · Authors · 2018-01-05
**Paper Revisions According to Reviews**

We thank the assigned and the volunteer reviewers for their comments. We have made a major revision of the paper to tone down the writing and to make it easier to understand. We have also clarified how our work relates to earlier studies of visual concepts to make our contribution clearer. We can release the code by January 19 to address the reproducing issue mentioned by the volunteer reviewer.

Besides the modifications of writing, we also updated our paper with an Appendix which is consisted of:

(1) Ablations studies that include the comparison between VC distance based model and VC-Encoding based model (which was mentioned by AnonReviewer3), comparison between K-Means clustering and vMFM based method (which was mentioned by AnonReviewer2), and the study on VCs of different scales (which is the question of VCs from different layers raised by AnonReviewer2).

(2) More visualization and statistics on VCs to better illustrate their properties.

---

> ### Author Response · Authors · 2018-01-12
> **Source Code (a primary version) Released**
>
> We have released a primary version of our source code. Please find it on https://github.com/Awcrr/FewshotVC. We will keep completing its documents.

---

### Decision · Program_Chairs · 2018-01-29
**ICLR 2018 Conference Acceptance Decision**

**Decision:**

Reject

**Comment:**

The paper builds on earlier work by Wang et al (2015) on Visual Concepts (VCs) and explores the use of VCs for few-shot learning setting for novel classes.

The work, as pointed out by two reviewers is somewhat incremental in nature, with main novelty being the demonstration of utilities of VCs for few shot learning. This would not have been a big limitation if the paper had a carefully conducted empirical evaluation providing insights on the effect of various configuration settings/hyperparameters on the performance in few shot learning, which two of the reviewers (Anon3, Anon2) state are missing. The paper falls short of the acceptance threshold in its current form.

PS: The authors posted a github link to the code on Jan 12 which may potentially compromise the anonymity of the submission (though it was after all the reviews were already in) https://openreview.net/forum?id=BJ_QxP1AZ&noteId=BJaIDpBEM